# The cation exchanger Letm1, circadian rhythms, and NAD(H) levels interconnect in diurnal zebrafish

Pauline Dao[1,2,3,*], Stefan Hajny[1,2,*] , Ronald Mekis[2,3], Lukas Orel[1] , Nora Dinhopl[4], Kristin Tessmar-Raible[1] , Karin Nowikovsky[2,3]

Mitochondria are fundamental for life and require balanced ion exchange to maintain proper functioning. The mitochondrial cation exchanger LETM1 sparks interest because of its pathophysiological role in seizures in the Wolf Hirschhorn Syndrome (WHS). Despite observation of sleep disorganization in epileptic WHS patients, and growing studies linking mitochondria and epilepsy to circadian rhythms, LETM1 has not been studied from the chronobiological perspective. Here we established a viable *letm1* knock-out, using the diurnal vertebrate *Danio rerio* to study the metabolic and chronobiological consequences of *letm1* deficiency. We report diurnal rhythms of Letm1 protein levels in wild-type fish. We show that mitochondrial nucleotide metabolism is deregulated in *letm1−/−* mutant fish, the rate-limiting enzyme of NAD$^+$ production is up-regulated, while NAD$^+$ and NADH pools are reduced. These changes were associated with increased expression amplitude of circadian core clock genes in *letm1−/−* compared with wild-type under light/dark conditions, suggesting decreased NAD(H) levels as a possible mechanism for circadian system perturbation in Letm1 deficiency. Replenishing NAD pool may ameliorate WHS-associated sleep and neurological disorders.

## Introduction

Mitochondria are vital organelles and key metabolic hubs that control cellular biosynthetic pathways, chromatin modification, immune responses, stem cell pluripotency, and dictate life and death decisions (Spinelli & Haigis, 2018; Martinez-Reyes & Chandel, 2020). Originating from symbiotic bacterial oxidative ATP donors, mitochondria have evolved into highly dynamic reticular networks that communicate with all other intracellular organelles to exchange substrates, transfer signals, and coordinate functions. Their best-studied role as ATP providers integrates a wide range of regulatory biological mechanisms including circadian rhythms (Aguilar-Lopez et al, 2020).

Chemiosmotic energy conservation is the key mechanism for oxidative ATP production and aerobic life. Synthesis of ATP by chemiosmosis occurs at the inner mitochondrial membrane (IMM) and results from concerted multistep metabolic activities that converge on the generation of an electrochemical gradient across the IMM, also known as proton motive force. The electrochemical gradient is driven by chemiosmotic coupling of free energy to the osmotic potential of proton concentration and depends on: (i) the impermeability of the IMM to protons and generally to ions (fourth postulate of the chemiosmotic hypothesis), (ii) the existence of reversible electroneutral OH$^-$-anion, H$^+$-cation, and substrate diffusion systems (third postulate of the chemiosmotic hypothesis) (Mitchell, 2011). Chemiosmotic energy transduction thus coordinates ATP phosphorylation with mitochondrial translocation of substrates and ions, according to metabolic needs and physiological functions.

Chemiosmosis is a critical source of ATP in high-energy consuming tissues, like the brain, which uses ATP to sustain neuronal proliferation, activity and connectivity. The primary fuel for ATP in the brain is glucose. A complex set of regulated oxidation–reduction reactions involving oxidized and reduced nicotinamide adenine dinucleotide (NAD$^+$/NADH) and Ca$^{2+}$ signaling processes, partitions glucose metabolism between cytosol and mitochondria and adjusts the balance between glycolytic and oxidative ATP (Yellen, 2018; Diaz-Garcia et al, 2021). In addition, the circadian dimension of glucose metabolism and ATP production drives wake and sleep cycles over day and night (Reinke & Asher, 2019). Consistent with rhythmic fluctuation of energy metabolism, diurnal ATP rhythmicity correlates with oscillating intracellular Ca$^{2+}$ dynamics and NAD$^+$/NADH ratios as rate-determinants for ATP production (Reinke & Asher, 2019). This observation raises the question of whether internal clocks also set the pace of mitochondrial solute carriers and H$^+$-cation exchangers, as these are critical for chemiosmotic processes. In this context, a correlative

[1]Max F Perutz Laboratories, Research Platform Rhythms of Life, University of Vienna, Vienna, Austria   [2]Department of Internal Medicine I, Medical University Vienna, Vienna, Austria   [3]Department of Biomedical Sciences, Unit of Physiology and Biophysics, University of Veterinary Medicine Vienna, Vienna, Austria   [4]Department of Pathobiology, Institute of Pathology, University of Veterinary Medicine Vienna, Vienna, Austria

Correspondence: kristin.tessmar@mfpl.ac.at; karin.nowikovsky@vetmeduni.ac.at
*Pauline Dao and Stefan Hajny contributed equally to this work.

study between diurnal fluctuation of cardiac OXPHOS and Ca$^{2+}$ dynamics has shown increased activity of the mitochondrial Na$^+$/Ca$^{2+}$ exchanger and Ca$^{2+}$ transporters, as well as a higher mitochondrial Ca$^{2+}$ retention capacity at night than in the day, and attributed the increased cardiac daytime vulnerability to reduced daytime mitochondrial Ca$^{2+}$ dynamics (Abdel-Rahman et al, 2021).

The mitochondrial H$^+$-cation exchanger LETM1/SLC55A3 is one of the members of the solute carrier family, a large family of metal, inorganic ion, vitamin, and neurotransmitter transporters. The unique features of LETM1 are its essentiality in mitochondrial regulation of volume and cation homeostasis and its link to the pathogenesis of seizures. LETM1 haploinsufficiency correlates with seizures in the Wolf Hirschhorn Syndrome (WHS) (Endele et al, 1999; Rauch et al, 2001; Nevado et al, 2020). In addition, studies reported reduced levels of LETM1 in epileptic mouse models (Zhang et al, 2014) and higher threshold to pharmacologically induced seizures in LETM1 hemizygous mice (Jiang et al, 2013). Therefore, understanding the mechanistic contribution of LETM1 to seizures is needed, especially in the light of the physiological role of volume regulation in the control of mitochondrial bioenergetics and motility, as well as in cognitive and behavioral processes.

LETM1 is a mitochondrial osmoregulator implicated in the mitochondrial K$^+$-H$^+$ and Ca$^{2+}$-H$^+$ exchange, a key function opposing the Nernstian equilibrium potential of K$^+$ and Ca$^{2+}$ that is imposed by the electrical chemical gradient (Austin et al, 2017; Austin & Nowikovsky, 2019). Furthermore, LETM1 is involved in a number of other biological functions. These include shaping the mitochondrial network and architecture of the inner membrane cristae (Piao et al, 2009; Nakamura et al, 2020), and functions related to mitochondrial biogenesis, like the assembly of respiratory complexes (Tamai et al, 2008), mitochondrial translation or mitochondrial DNA organization (Durigon et al, 2018). LETM1 is also critical for glucose metabolism. In the brain of letm1(+/−) mice, an increase of glycolytic metabolites and a reduced activity of the Ca$^{2+}$-sensitive pyruvate dehydrogenase (PDH), the first enzyme that commits cells to glucose oxidation was detected, as well as decreased metabolites of the de novo biosynthesis of NAD$^+$, tryptophan, and quinolate and reduced levels of ATP (Jiang et al, 2013). However, whether NAD$^+$ stores are altered under LETM1 deficiency has not been investigated.

To explore the metabolic and circadian activities potentially perturbed by LETM1 deficiency, we used the zebrafish as a diurnal vertebrate model system. Making use of gene editing technologies, we established a new knock-out model to study the deletion phenotypes of zebrafish letm1 and uncover new aspects of letm1 pathogenicity.

# Results

## Dre-Letm1 exists as a highly conserved single orthologue with particularly high transcript levels in the brain

Members of the LETM1 protein family are evolutionarily conserved throughout all sequenced eukaryotes. All members share a protein architecture characterized by a highly preserved membrane-spanning region and LETM-like domain annotated as PFAM PF07766, and, except for HCCR1/LETMD1 proteins (Cho et al, 2007), a submitochondrial protein distribution at the inner membrane. Despite the common genome duplication in the teleost zebrafish lineage, there is only one LETM1 orthologue in Danio rerio, letm1, also annotated as si:ch211-195n12.1 or si:rp71-77d7.1., and located on chromosome 13. Zebrafish Letm1 protein is highly conserved and shares 65% identities with human LETM1 and 64% with mouse LETM1. We generated a phylogenetic tree to evolutionarily cluster vertebrate, invertebrate, plant and yeast LETM1 protein family members. HCCR1/LETMD1 was used as the out-group. Like all other vertebrates, zebrafish Letm1 has a conserved vertebrate orthologue Letm2, whereas a single letm1/2 gene exists in invertebrates (Fig 1A).

To evaluate the possible functional conservation of zebrafish letm1 to human LETM1, we expressed a GFP-fused version of letm1 in HeLa cells. We also stained HeLa cells with the live mitochondrial specific indicator MitoTracker Red (MTR). The overlay of Letm1-GFP and MTR fluorescence illustrated the mitochondrial localization of ectopically expressed Letm1 in this mammalian cell line (Fig 1B–D) and suggested a conserved subcellular distribution from zebrafish to humans.

letm1 transcript levels showed ubiquitous expression during early developmental stages with an increased concentration in the head (Fig 1E). The broad expression in all organs and the enrichment of letm1 was also observed in adult fish brain, but also eyes and gonads, as revealed by qRT-PCR (Fig 1F). Similarly, whole-mount in situ hybridization in embryos at 1 day post fertilization (dpf) showed a higher expression in the head (Fig 1G and H). Because letm1 was more highly expressed in the head and brain, we performed in situ hybridization in the adult brain to look at the distribution of the expression within the adult brain. The in situ showed strong staining of letm1 in the telencephalon, along the medial dorsal zone of dorsal telencephalon, and the dorsal and ventral nucleus of ventral telencephalon, and in the mesencephalon at periventricular gray zone and dorsal zone of periventricular hypothalamus (Fig 1I–O).

## Transcription activator-like effector nuclease–mediated premature stop codon leads to depletion of Letm1 protein

To study letm1 deficiency in zebrafish as an animal model, we generated letm1 mutants by taking advantage of the transcription activator-like effector nuclease (TALEN) technology. The mutant allele (a deletion of 16 bp, that is, Δ16) is predicted to result in a premature stop codon, leaving only 8 AA N-terminally (Fig 2A). Genotyping via PCR analysis of genomic DNA isolated from 2 dpf fish allowed to distinguish bands of different sizes: wild-type (349 bp) and homozygote (333 bp) fragments. Heterozygous fish showed both bands (349 and 333 bp). cDNA isolation and sequencing confirmed a 16 bp deletion within the first exon leading to a frame shift mutation and premature stop codon: c.Δa19-c34fs6*/p. ΔT6-L13 (Fig 2A). To determine the Letm1 protein status in letm1−/− mutants, we generated a monoclonal antibody against the zebrafish Letm1 using a C-terminal fragment (AA 462-768) for immunisation. Indicating that we very likely obtained a complete functional null mutant (Fig 2A), we confirmed the total absence of

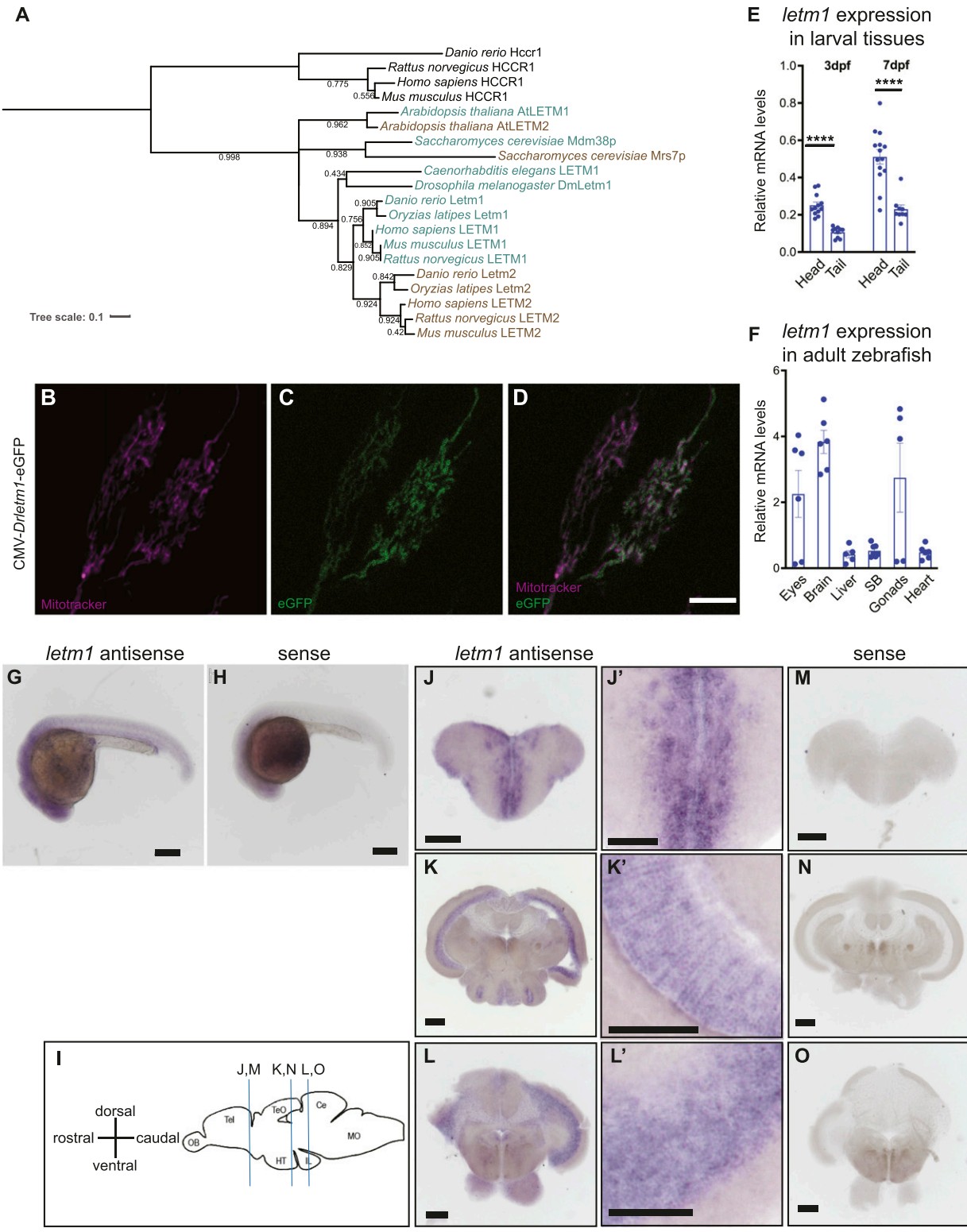

**Figure 1. The highly conserved gene *letm1* is broadly expressed in zebrafish with high levels in brain. (A)** Phylogenetic analysis of Letm1/2 family proteins using the free online access tools phylogeny.fr and itol.embl.de/. HCCR1 was used as an out-group. **(B, C, D)** Confocal images of HeLa cells expressing CMV-*Drletm1*-eGFP (green) and stained with MitoTracker Red (magenta). Scale bar 10 μm. **(E)** Relative mRNA levels of *letm1* in zebrafish larvae, head, and tail at 3 and 7 day post fertilization (dpf), normalized to the *eukaryotic elongation factor-1 α* (*eef1a*). Each data point represents a biological replicate. 3 dpf head (n = 12); 3 dpf tail (n = 9); 7 dpf head (n = 14); 7 dpf tail (n = 9). *t* test: ****$P \leq 0.0001$. **(F)** Relative mRNA levels of *letm1* in adult organs (SB, swimming bladder), normalized to *eef1a*. Each data point represents a biological replicate. Mean ± SEM. **(G, H)** In situ hybridization of *letm1* on wild-type embryos at 1 dpf. Scale bar 1 mm. **(I)** Schematic representation of the adult zebrafish brain in lateral

Letm1 protein in zebrafish homozygous mutant larvae by Western blot (Fig 2B).

### Mitochondrial localization and diel oscillation of zebrafish Letm1

Given the strong conservation in sequence and localization across species (Fig 1B–D), we next wanted to validate the mitochondrial localization of Letm1 in zebrafish, and performed immunofluorescence in zebrafish cells using our *Dre*-Letm1 antibody. Confocal microscopy confirmed the exclusive mitochondrial localization of Letm1 and strongly suggests a functional conservation of the protein in the membrane of the mitochondria (Fig 2C–E). Immunostaining also allowed us to independently confirm the absence of Letm1 in zebrafish *letm1–/–* cells (Fig 2F–H).

More recent evidence indicates that many cation channels can be regulated over daily time (Abdel-Rahman et al, 2021). We thus wondered whether the protein levels of Letm1 might change across the day. We sampled 6 dpf and 7 dpf wild-type larvae at ZT5, ZT11, ZT17, and ZT23 under a 16:8 white light/dark regime and extracted protein. Western blotting analysis revealed statistically significant changed Letm1 protein levels over daily time with a peak at ZT23 and a trough at ZT5 (Figs 2I and J and S1A–D), indicating a regulation of Letm1-depending physiology across the day. The establishment of physiological rhythms at around 4–5 dpf (Ben-Moshe et al, 2014) may explain the increasing robustness of diel Letm1 oscillations. These rhythmic changes were not detected at the transcriptional level (Fig S2) suggesting diel protein level regulation. This is consistent with previously reported oscillating posttranslational modifications associated with a weak correlation between mitochondrial transcriptome and proteome (Reinke & Asher, 2019).

### The homozygous *letm1* mutation leads to high embryonic lethality, but survival of a small population of compensatory offspring

Given the specific spatial expression and temporal regulation of *letm1*/Letm1, we next aimed for the characterization of its function in our zebrafish mutants. As expected from analyses in other animal models including mice and flies (McQuibban et al, 2010; Jiang et al, 2013), we observed that the embryonic development was severely altered in *letm1–/–* fish. This maldevelopment was associated with a mortality of about 70% of embryos that did not survive longer than 24 h post fertilization and survivors with severe forms of malformation (Fig 3A–D). However, unexpectedly, we also obtained a small proportion of phenotypically normally developing fish that even reached adulthood and were breedable (Fig 3B–D). This finding suggested that a potentially interesting compensatory mechanism can be activated in a minority of the mutants.

### Abnormal mitochondrial morphology in *letm1*–deficient larvae

The phenotypically wild-type *letm1–/–* survivors are from here on called compensators. These compensators now allowed us to study physiological consequences of *letm1–/–* that are completely missed in non-survivors or heterozygous animals and also to better understand which pathways need to be regulated to ensure survival despite the lack of this important gene.

The main feature of LETM1 is its role in keeping ion homeostasis and thus, the volume of the mitochondria. We therefore next addressed the level of changes in the mitochondrial morphological in *letm1–/–* larvae, both non-compensating, as well as compensating for the deficiency. Transmission electron microscopy (TEM) was used to analyze the mitochondrial ultrastructure of *letm1–/–* in comparison with *letm1+/+*. *letm1–/–* larvae were divided into the two groups of malformed (non-compensators) and healthy (compensators) developers. Mitochondria in skeletal muscle fibers at 4 dpf are not yet fully organized; however, we observed densely packed *letm1+/+* mitochondria in the sarcomeres. *letm1–/–* mitochondria from malformed larvae were scarce, more damaged, smaller with a detached outer membrane (Fig 3E, F, and F'). The cristae were more blurred and were often partly filled with empty shapes resembling vacuoles. In contrast, mitochondria of *letm1–/–* healthy developing larvae exhibited an intermediate morphology between the strongly malformed *letm1–/–* and the healthy *letm1+/+* controls, displaying fewer damaged or empty mitochondria (Fig 3E–G'). The altered morphological phenotype suggests that the compensatory mechanism of the surviving larvae already occurs at the mitochondrial level.

### NAD⁺ homeostasis is disturbed in the absence of *letm1*

We next wondered about the physiological functionality of *letm1–/–* animals. Changes in tryptophan and quinolinic acid, the main components for NAD biosynthesis, had been pointed out by metabolites analysis in *letm1+/–* mice, but never further investigated (Jiang et al, 2013). Following up on this, we investigated whether *letm1* deficiency affects the NAD(H) pool. We measured $NAD^+$ and NADH content in embryos at the early developmental stage of 1 dpf (compensators and non-compensators combined). $NAD^+$ and its reduced form NADH were both decreased in *letm1–/–* embryos compared with *letm1+/+* embryos (Fig 4B). Whereas the decrease was significant for NADH, a clearly similar trend was observed for $NAD^+$. The $NAD^+$/H ratio of 1 dpf embryos was unexpectedly low compared with mammals. One explanation for this could be that embryonic fish at the age of 1 dpf still have a predominant glycolytic metabolism that favors NADH pools, possibly consistent with the presence of many dividing, undifferentiated cells.

Although $NAD^+$/NADH cycling is crucial to sustain cellular energy conservation, $NAD^+$ is also consumed by enzymes during DNA repair, deacetylation, and ADP ribosylation processes and inflammatory

---

view. Blue vertical lines represent the location of the brain in lateral view. Blue vertical lines represent the location of the brain slices displayed. OB, olfactory bulb; Tel, telencephalon; TeO, optic tectum; Ce, corpus cerebelli; HT, hypothalamus; IL, inferior lobe; MO, medulla oblongata. **(J, K, L)** In situ hybridization of *letm1* on coronal slices of wild-type adult brain. **(J', K', L')** Magnification of the left panel. **(M, N, O)** Sense control. Scale bars 500 *μm*.

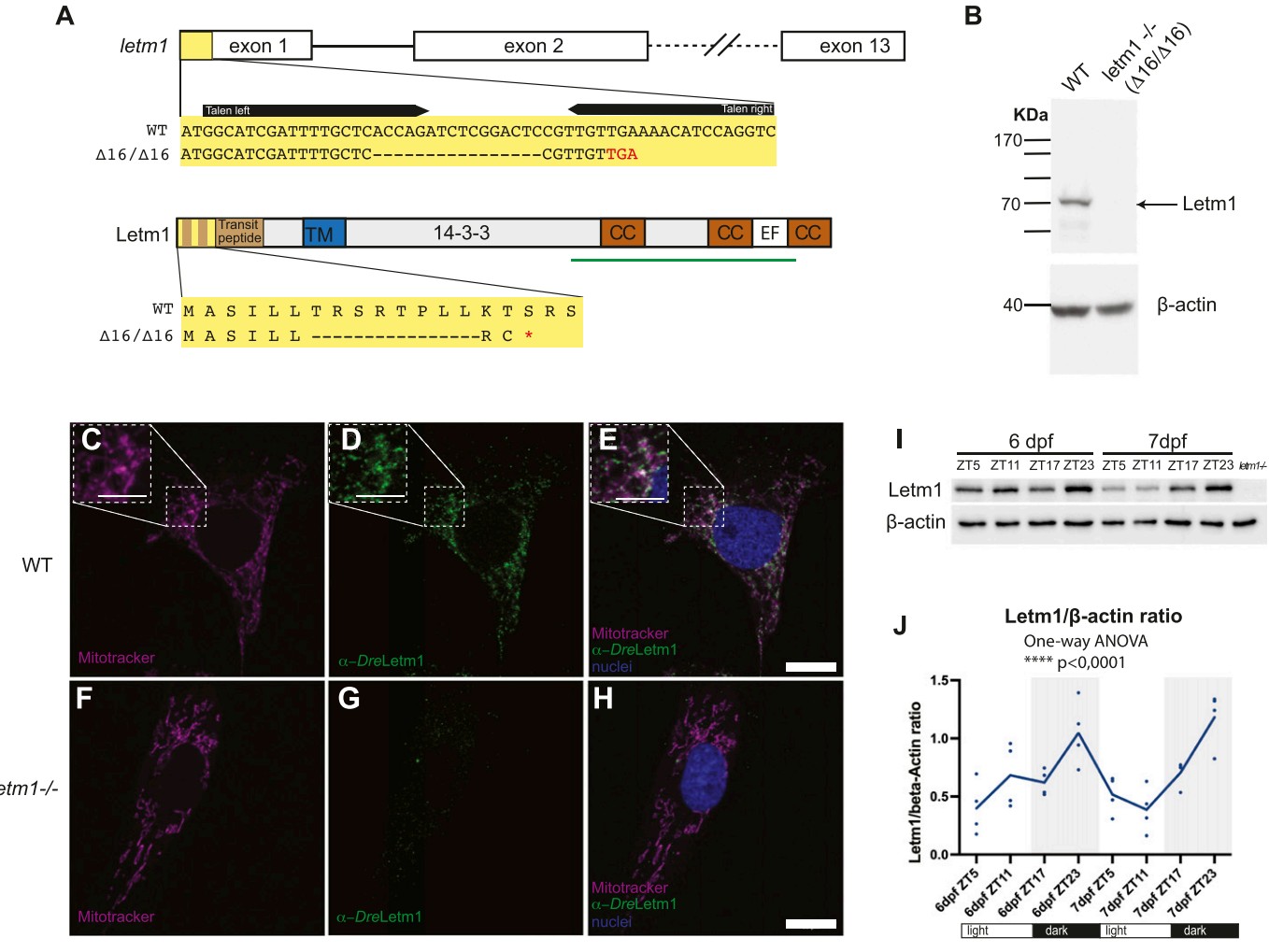

**Figure 2. Letm1 localizes exclusively in mitochondria and displays a diel rhythm. (A)** Schematic representations of *letm1* gene and protein below with the locus (yellow box) of the deletion induced by the TALEN pairs. The region deleted (Δ16) is represented in "–." The red asterisk represents a stop codon. TM, transmembrane region. 14-3-3, also known under the name of LETM Domain or LETM1 RBD (ribosomal binding domain) CC, coil–coil; EF, putative $Ca^{2+}$-binding EF hand motif. Green bar indicates the protein fragment used to generate the *Dre*-Letm1 antibody. **(B)** Immunoblot using 6 day post fertilization zebrafish protein lysates from WT and *letm1*–/– with β-actin as a loading control. **(C, D, E, F, G, H)** Confocal images of WT and *letm1*–/– zebrafish cells immunolabeled with zebrafish anti-Letm1 (green) and stained with MitoTracker Red (magenta) and DAPI (blue). Scale bar 10 μm. **(I)** Representative immunoblots using 6 and 7 day post fertilization zebrafish protein lysates from WT and *letm1*–/– fish probed with anti-Letm1, anti-Tom20 and anti–β-actin. **(J)** Quantitative graph of four biological replicates (see Fig S1A–D) representing the relative protein level of Letm1 to β-actin over circadian time points. One-way ANOVA, n = 4.

responses. Because of NAD+ degradation by consumption, cells must continuously replenish their NAD+ pools, either by de novo synthesis from the tryptophan derivate quinolinic acid or by salvage pathways from recycled precursors of the vitamin niacin, such as nicotinamide (NAM). Nicotinamide phosphoribosyltransferase (NAMPT), the rate-limiting enzyme of the NAD+ salvage pathway is an evolutionary ancient molecule (Fig 4A) that coordinates glucose availability and NAD+ levels. NAMPT gene duplication occurred at the base of vertebrates, resulting in *namptA* (*nampt1*) and *namptB* (*nampt2*) (Fang et al, 2015). Of those groups, teleost fish retained both groups, whereas air-respiring vertebrates have just retained NAMPTA. It is thought that *nampt2* correlates with the decrease in oxygen levels during the water-to-land transition (Fang et al, 2015). To assess the expression of *nampt* in *letm1*–/–, we performed qRT-PCR analysis of *nampt1* and *nampt2* over diel timepoints in 6 dpf *letm1*+/+ and *letm1*–/– larvae (Fig

4C and D). In control fish, *nampt1* exhibited rhythmic expression, whereas *nampt2* did not. *nampt1* showed similar diel changes in *letm1*–/–, but with a significantly elevated expression amplitude at ZT5. In contrast to *letm1* +/+ larvae, *nampt2* markedly peaked at ZT5 in *letm1*–/– larvae. We interpret this robust increase in the way that NAD+ pools were exhausted after the dark phase, possibly through oxygen-glucose shortage or increased NAD+ consuming enzymatic reactions and that *nampt* genes become up-regulated to compensate for NAD+ depletion.

**The *letm1* deficiency results in diel amplitude changes of core circadian clock transcripts**

Given the much more prominent diel regulation of *nampt1* and *nampt2* in *letm1*–/– fish at 6 dpf, and NAD+'s central position in the

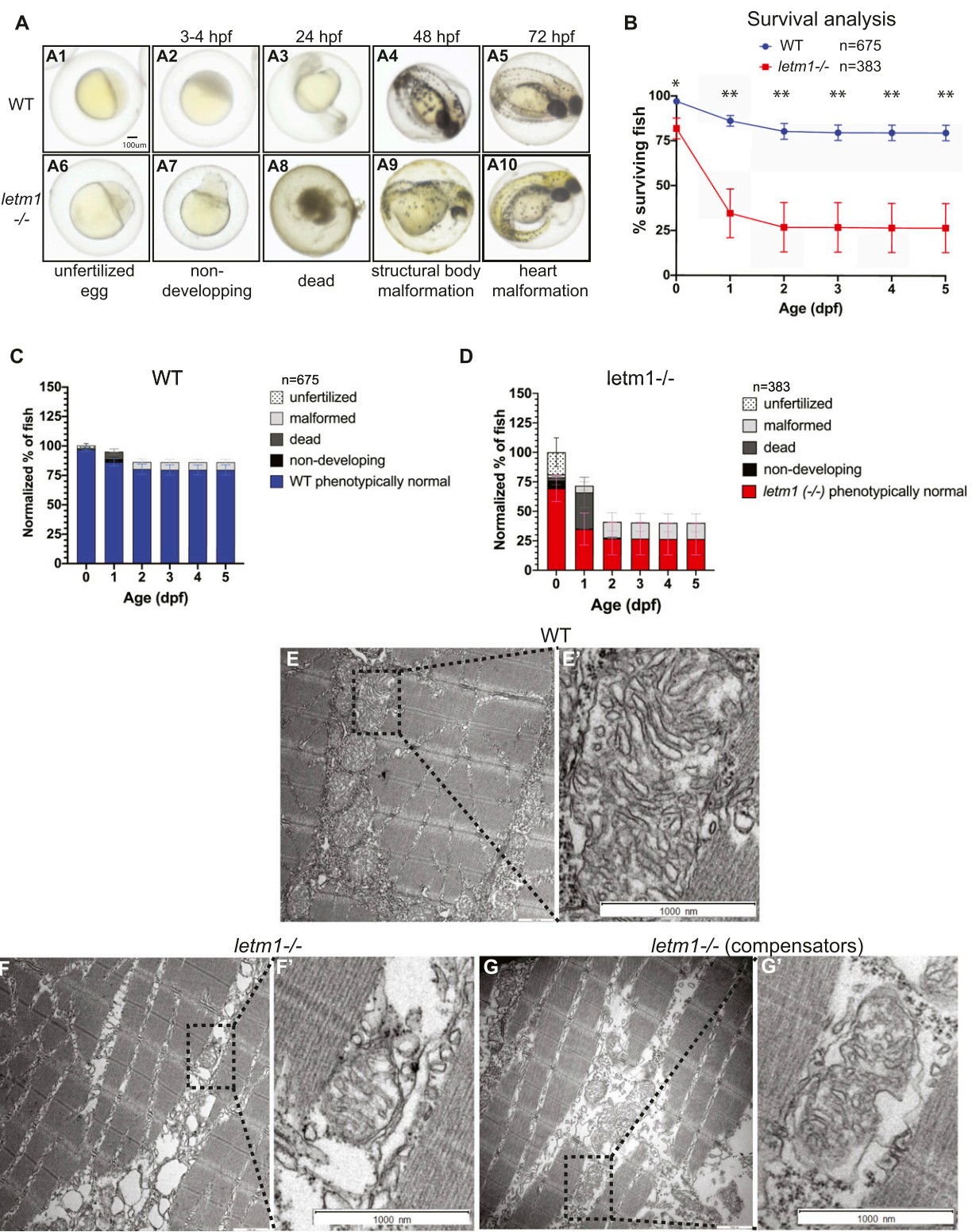

**Figure 3. *letm1* KO impacts embryonic development, survival and mitochondrial morphology. (A)** Images of WT and *letm1−/−* zebrafish embryos. Lower panel display representative images of the different categories used in the phenotypic analysis. **(A1–A5)** Representative images of wild-type, phenotypically normal fish. **(A6–A10)** Representative images of *letm1−/−*, (A6) unfertilized egg, (A7) non-developing, (A8) dead, (A9) structural body malformation, and (A10) heart malformation. Scale bar: 100 μm. **(B)** Survival analysis of zebrafish wild-type (WT) and *letm1−/−* from 0 to 5 day post fertilization (dpf). Quantification of the number of surviving fish per day shown in percentage. The total number of fish analyzed was $n_{WT}$ = 675 and $n_{letm1−/−}$ = 363. Graph shows mean ± SEM. Welch *t* test, *$P < 0.05$. **(C, D)** Phenotypic analysis of zebrafish WT (C) and *letm1−/−* (D) from 0 to 5 dpf. **(E, F, G)** Transmission electron microscopy images of 4 dpf zebrafish larvae: WT (E) Longitudinal section of muscle fibers show

feedback loop that coordinates *sirt1* and circadian clock gene regulation (Fig 4I) (Sassone-Corsi, 2012; Levine et al, 2020), its reduced levels made us wonder about changes in circadian core clock transcript levels, possibly explaining the increased diel amplitude changes of *nampt1* and *nampt2* transcripts. Thus, we analyzed the expression of *clock1a*, *per1b*, *per2*, and *bmal1a* at 6 dpf (Tamai et al, 2004; Dekens & Whitmore, 2008). Highly reminiscent of the increased diel transcript amplitudes of *nampt1* and *nampt2* in *letm1−/−* animals, qRT-PCR analyses revealed a consistently increased amplitude of all tested core circadian genes. This was statistically significant for *clock1a* at ZT17 (11 PM), *per1b* expression at ZT5 (noon), and ZT23 (6 AM) and *per2* at ZT5, that is, at their respective natural maximum (Fig 4E–G), and as a trend visible also for *bmal1a* (Fig S3A).

The observed reductions in NAD$^+$/NADH levels in *letm1−/−* mutants occur already at 1 dpf, a developmental time point at which the core circadian clock of the fish is still maturing (Dekens & Whitmore, 2008; Lahiri et al, 2014). We thus reason that the resulting core circadian clock amplitude changes are likely a consequence of the altered NAD$^+$/NADH levels. In turn, once initiated those increased core circadian clock amplitude changes might cause the observed diel *nampt1* and *nampt2* amplitude changes, possibly as part of the compensatory mechanism allowing some *letm1−/−* fish to survive.

## Genetic compensations for *letm1* deficiency

Collectively, loss of *letm1* pointed to early developmental lethality accompanied with severe malformations (Fig 3A), except for a limited percentage of escapers. Remarkably, these escapers/ compensators did not differ morphologically from wild-type larvae (Fig 5A and B), in contrast to the non-escapers (Fig 5C and D). To gain insight into possible compensatory survival changes on the molecular level from an unbiased perspective, we performed differential expression analysis using quantitative RNA sequencing data from 4 dpf *letm1+/+* (WT), *letm1−/−* compensator (C), and *letm1−/−* non-compensator (NC) larvae and performed a Go-term enrichment analysis (Fig 5E–G).

Consistent with our analyses of NAD$^+$/NADH analyses, we found changes in the tryptophan-kynurenine pathway (Fig 5H and I). This catabolic pathway is crucial in the brain and heart and not only serves de novo NAD$^+$ synthesis but also produces intermediate metabolites with neuroactive properties and links the immune system to the neurological system (Wang et al, 2020). Genes of this pathway were significantly decreased under *letm1* deficiency in both, compensators and non-compensators, but to a significantly higher extent in the latter. These data suggested a reduced availability of de novo NAD$^+$ and kynurenic acid.

Of note, pairwise analysis of compensators or non-compensators versus wild-type or versus non-compensators excluded transcriptional compensation by *letm2 or letmd1* (Fig S3B

and Tables S1–S3). The few notable changes in transcription patterns for K$^+$ transport systems concerned the plasma membrane voltage-gated K$^+$ transporters, with decreased *kcnj12a* in non-compensators and increased *kcnh6* in compensators. However, mitochondrial K$^+$ or Ca$^{2+}$ transporters were not differentially expressed in the three groups, except for the mitochondrial Ca$^{2+}$ uniporter *mcu* that was down-regulated in non-compensators (Table S3).

Among the solute carriers (SLCs), we found remarkable differences between compensators and non-compensators involving both plasma membrane and mitochondrial transporters (Fig S4). Transcripts for plasma membrane SLCs that regulate intracellular pH or volume (bicarbonate ions, protons, or inorganic ions transporters), or transport fatty acids were decreased in non-compensators only. Increased were transcripts for SLCs transporting glucose, monocarboxylate, phosphate, peptides, and organic ions (Fig S4A). In contrast to these exclusive changes in non-compensators, compensators decreased transcripts encoding Na$^+$-dependent glucose or sulphate/carboxylate, and neurotransmitter SLCs. Of the notable changes for specific members of the mitochondrial subfamily SLC25 (Fig S4B), the transcripts of *slc25a22a*, the glutamate-aspartate transporter that participates in the malate-aspartate shuttle providing cytosolic NADH to mitochondria, and those of the carnitine carrier (*slc25a20*) were the most reduced in non-compensators; in compensators, they were those of the uncoupler *slc25a7/ucp1*. The most elevated transcripts within non-compensators were for *slc25a5/ucp3*, an uncoupler family member with protective functions against oxidative damage, and *slc25a38a*, the orthologue of human *APOPTOSIN*. In the compensators it was the Ca$^{2+}$-dependent aspartate/glutamate transporter *slc25a13*.

Unexpectedly, the Gene Ontology analysis pointed to a different transcriptional landscape between compensators and non-compensators in biological processes related to innate immune and inflammatory responses (Figs 5E–G and S5A and B). Innate immune and inflammation genes were not significantly up-regulated and rather repressed in compensators versus wild-type. Yet, there was clearly differential expression between compensators and non-compensators, mainly due to the induction of chemokines, cytokines and receptors in the non-compensators. The significantly enriched genes in the latter belonged to the inflammatory clusters of the pore attack, matrix complement, and wound healing. In this connection, leptin genes, known to be self-enhancing cytokine-induced pro-inflammatory adipokines, were prominently up-regulated in non-compensators compared with compensators. Inflammatory and wound healing events interconnect with ECM formation and remodeling for tissue repair and regeneration. ECM genes encoding collagen, proteoglycan, fibronectin, laminin, hyaluronan formation, and metallopeptidases were however strongly repressed in the non-compensators (Figs S5 and S6B). Phenotypically wild-type–like compensators (Fig 5A and B) did

---

dense mitochondrial; malformed *letm1−/−* (F) have less cristae in intramyofibrillar mitochondria, arrow points to detached outer membrane, asterisk to a vacuole-like shape, possibly indicating the transition between damaged, and degraded mitochondria; *letm1−/−* compensators, with denser mitochondria than in non-compensator *letm1−/−* and apparently not fusing with vacuoles (G). Scale bar 1,000 nm. **(E′, F′, G′)** Magnification of the boxed dotted line area.

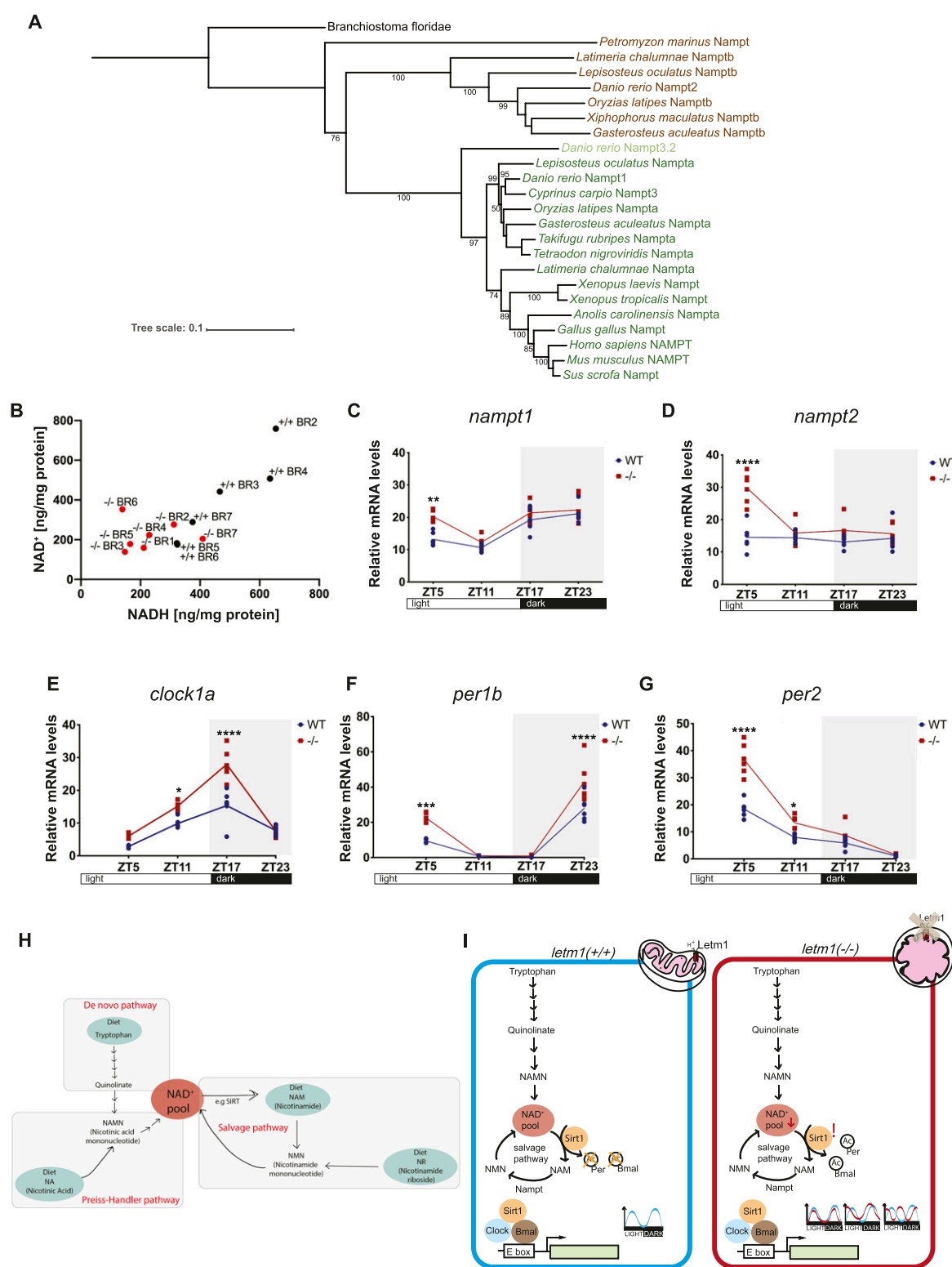

**Figure 4. Evolutionary tree of Nampt, measurement of NAD⁺/H levels, and expression of *nampt* and clock genes. (A)** Phylogenetic analysis of Nampt protein. **(B)** Measurement of NAD⁺ and NADH in 1 day post fertilization (dpf) zebrafish embryos WT and *letm1−/−*. Each data point represents a biological replicate, each biological replicate contained 20 embryos. N(WT) = 6 and n(*letm1−/−*) = 7 biological replicates, NAD⁺ $P$ = 0.0789, NADH *$P$ < 0.05, Holm multiple comparison. **(C, D)** Relative mRNA levels across circadian time points of *nampt1* and *nampt2* in 6 dpf zebrafish larvae WT (blue) and *letm1−/−* (red). The expression is normalized to *β-actin*. Each data point represents a biological replicate, n = 6 biological replicates. One-way ANOVA with Tukey's multiple comparisons test: ****$P$ < 0.0001. **(E, F, G)** Relative mRNA levels of *clock1a* (E), *period1b* (per1b) (F), and *period2* (per2) (G) normalized to *β-actin* in 6 dpf larvae. Each data point represents a biological replicate, n = 6. One-way ANOVA with

not differentially express extracellular collagen formation genes (Fig S6B). Given that compensators and non-compensators used for sampling showed no visible signs of tissue damage or necrosis (Fig 5A–D), it is conceivable that the observed immune/inflammatory response changes might reflect a molecular signature of the non-compensators that contributes to their survival problems, rather than just a signature of tissue decay. Of note, several genes up-regulated in the non-compensators are involved in metabolic redox reactions, cholesterol biosynthesis, and lowering resistance to oxidative stress (Fig S6A). This may indicate a higher metabolic and oxidative stress in the non-compensators compared with wild-type and compensators. Overall, gene enrichment analysis might suggest a potential link between survival and control of the immune response.

# Discussion

We generated a zebrafish model for *letm1* deficiency and showed that homozygous deletion of *letm1* causes a high rate of embryo lethality, but can also be compensated for by a yet unknown mechanism. Surviving larvae developed into larva and breedable, adult fish with no apparent phenotype. This survival phenotype of "healthy" compensators that we observed in zebrafish was not found in flies, mice, or humans. An explanation for obtaining compensator individuals in zebrafish is possibly the much high number of offspring. Similar compensators might be possible in mice and humans, but as they are rare would not be detected by the low offspring numbers of mice and humans. Flies arguably produce high numbers of offspring. However, they possess only a single *letm1/2* gene, instead of the distinct *letm1* and *letm2* genes in vertebrates. This implies that the fly mutant rather corresponds to a double knock-out of *letm1/letm2* in vertebrates, which might explain why no compensators were detected in flies. We consider the uncovered *letm1–/–* survivors in zebrafish as an interesting advantage that can be exploited in future studies to uncover the underlying survival processes to obtain a deeper understanding on the plasticity of the process. Micrographs revealed a positive correlation of lower mitochondrial defects with the developmental rescue, suggesting mitochondrial compensatory effects in the survivors, most likely by balanced NAD$^+$/H pools through differential expression of mitochondrial SLCs and biosynthetic and salvage factors, and supporting the notion of mitochondrial dysfunction in the etiology of LETM1-related diseases and seizures.

The particularly high abundance of *letm1* in different parts of the zebrafish brain will likely be of future interest in the context of LETM1-linked behavioral abnormalities in humans. The mechanism linking LETM1 and seizures in WHS or temporal lobe epilepsy is, however, still poorly explored. Some therapeutic successes in the treatment of seizures in WHS have been achieved by targeting voltage-gated Na$^+$ and Na$^+$-dependent Ca$^{2+}$ channels (Motoi et al, 2016). Regardless of the debate as to whether K$^+$ or Ca$^{2+}$ is exchanged for H$^+$ by LETM1, LETM1 deficiency most likely indirectly affects intracellular Ca$^{2+}$ homeostasis and mitochondrial NCLX activity (Austin et al, 2017).

In this study, we did not focus on *letm1*-mediated cation homeostasis but instead used the zebrafish as a diurnal vertebrate to approach the role of *letm1* from a different angle. We discovered fluctuating levels of Letm1 in a diurnal rhythm with a peak at ZT23. We showed that *letm1* depletion decreases intracellular NAD$^+$ and NADH. The fact that this phenotype occurs at the early developmental stage (1 dpf), where surviving and non-surviving *letm1–/–* offspring are indistinguishable, and that, in contrast, the diel expression of *nampt1/2* is up-regulated in surviving 6 dpf *letm1–/–* larvae, which are exclusively compensators, supports the vital role of NAD(H) and the compensatory vital function of *nampt1/2*. This regulatory logic lets us propose a central position of *nampt* in a self-reinforcing feedback loop between the regulation of clock genes by Sirt1 through the consumption of NAD$^+$, the periodic expression of *nampt* genes by *clock1a* and *bmal1* through their binding to the regulatory E-boxes of *nampt*, and the oscillation of NAD$^+$ supply by *nampt* (Nakahata et al, 2009) that ultimately results in viable NAD(H) levels (Fig 4H and I).

The higher NAD$^+$/NADH ratio in *letm1–/–* compared with *letm1+/+*, due to the stronger decrease in NADH relatively to NAD$^+$, can have several explanations: higher oxidation rate of NADH through respiratory complex I and aerobic NAD$^+$ regeneration, increased conversion of pyruvate to lactate and hypoxia-like glycolysis, or a higher dependence on ketone bodies (KB). Indeed, ketone metabolism also depends on the NAD(H) pool, with the NAD$^+$/NADH ratio determining whether KB are oxidized or formed. LETM1-deficient WHS fibroblasts were found to rely on KB oxidation as a source of acetyl-CoA to compensate for defective pyruvate dehydrogenase (Durigon et al, 2018). KB utilization instead of glucose to provide acetyl-CoA uses less NAD$^+$ than glucose, consequently more cytosolic NAD$^+$ remains available for NAD$^+$-consuming enzymes (Newman & Verdin, 2017; Xin et al, 2018). The reported antiepileptic effect ketogenic diets could very well be linked to the sparing use of NAD$^+$.

The central role of NAD$^+$/H in *letm1* was supported by our quantitative transcript level analyses. Mitochondrial NAD$^+$ is crucial for the enzymatic activity of the TCA cycle and the supply of fuels for oxidative ATP production. Although the mitochondrial NAD$^+$ carrier *slc25a51* was similarly up-regulated in both *letm1–/–* groups, *slc25a22a*, the aspartate-glutamate transporter orthologue, which is involved in the malate-aspartate shuttle and cytosolic NADH translocation into mitochondria, appeared strongly down-regulated in non-compensators only. Together, with decreased transcripts of *pyruvate dehydrogenase E1*, the entry point of pyruvate into the TCA cycle, in non-compensators only, it is possible to speculate a role of reduced NAD$^+$/H bioavailability in non-compensators in reduced survival.

We also show that the oscillation amplitudes of circadian clock genes are elevated in *letm1* deficiency in a diurnal vertebrate. Given the association of sleep disorganization in epileptic WHS (Battaglia et al, 2003) and the association of altered clock gene expression with seizure susceptibility, it is suggested that diurnal rhythmicity is

Tukey's multiple comparisons test: *P < 0.05; ***P < 0.001 ****P < 0.0001. **(H)** Schematic summarizing the NAD$^+$ production pathways in the cell, adapted from Verdin (2015). **(I)** Schematic of a proposed model linking the effect seen on the circadian clock, based on Sassone-Corsi (2012) and Levine et al (2020).

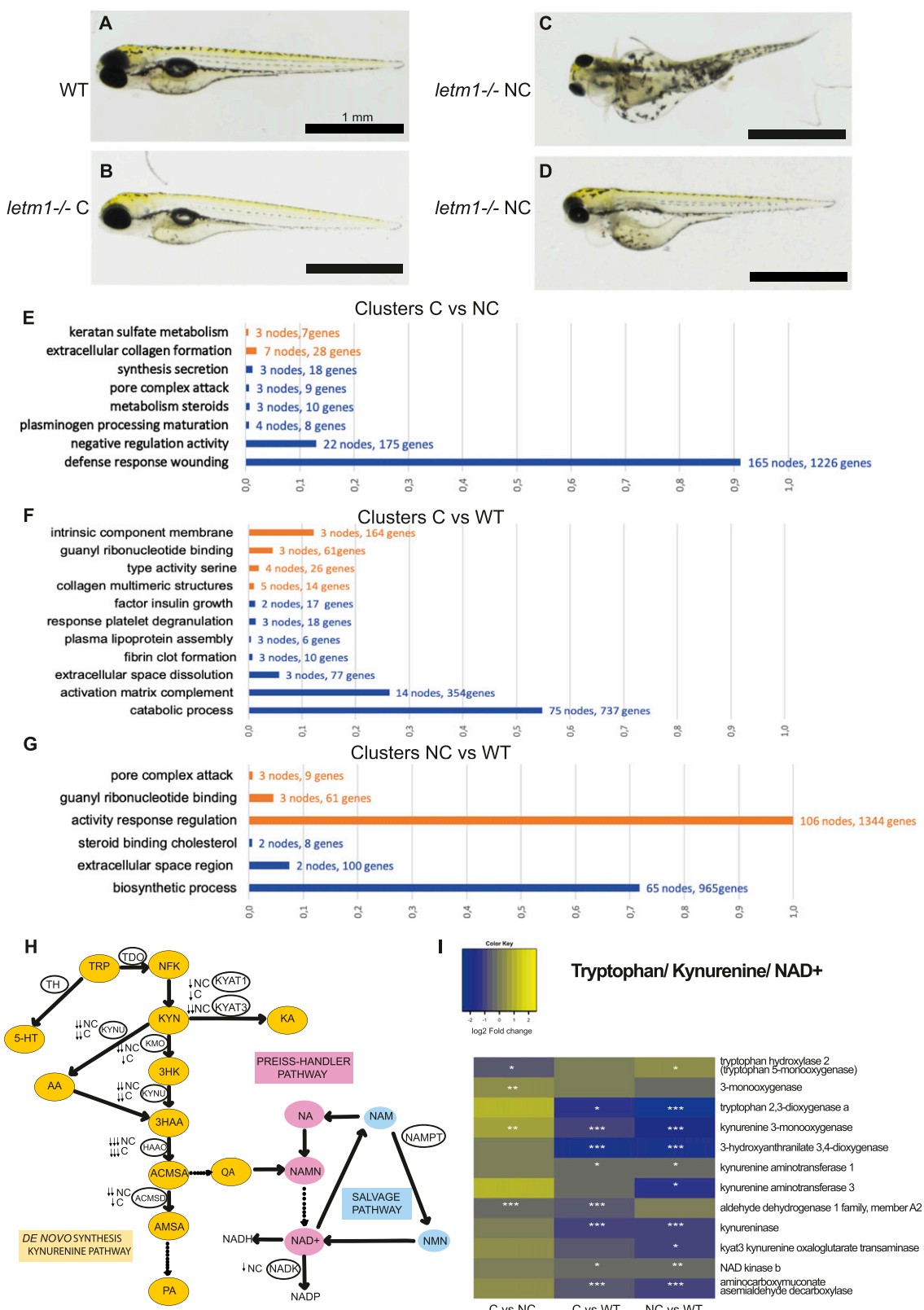

**Figure 5. Comparison of *letm1+/+* and *letm1−/−* compensators versus non-compensators. (A, B, C, D)** Representative images of zebrafish larvae at 4 day post fertilization, wild type (WT) (A), *letm1−/−* compensators (C) (B), and *letm1−/−* non-compensators (NC) (C, D), sampled for RNA sequencing. Scale bar = 1 mm. **(E, F, G)** Gene set enrichment analysis grouped in clusters according to gene ontology terms. **(E, F, G)** Up-regulated clusters (orange) and down-regulated clusters (blue) are shown for each pairwise comparison C versus NC (E), C versus WT (F), and NC versus WT (G). The threshold was set to *P* (adjusted) value = 0.05 and log$_2$ fold change = ±1. **(H)** Schematic

**TALEN pairs targeting *letm1* Exon 1**

| Letm1-Exon1 | |
| --- | --- |
| Nucleotide sequence of the left TALEN array | 5′-GGCATCGATTTTGCTCACC-3′ |
| Nucleotide sequence of the right TALEN array | 5′-GACCTGGATGTTTTCAACAA-3′ |
| Restriction enzyme for screening purposes | PleI (#R0515; NEB) |

involved in the development of seizures (Chan & Liu, 2021). In this context, our findings might also provide a new perspective on the contribution of *letm1* to seizures. We propose decreased NAD(H) pool as molecular link between perturbations of diurnal gene oscillations and seizure predisposition in the context of *LETM1* haploinsufficiency or homozygous *LETM1* variants.

Finally, we used compensator and non-compensator progeny to investigate potential survival mechanisms at the transcriptional level. Our data exclude transcriptional compensation via down or up-regulation of mitochondrial cation uptake or release pathways, respectively, or functional replacement of *letm1* in mitochondrial cation and volume homeostasis by other family members. Of course, compensation by posttranslational modifications cannot be excluded. Nevertheless, data clearly pointed to the unexpected consequence of *letm1* deficiency for the immune system that go well beyond the mitochondrial or cellular level. More specifically, our data suggest that a compensatory mechanism is critical to keep the immune system under control. Whereas significantly changed on multiple levels in the non-compensators, the compensators (which are morphologically indistinguishable from wild-type) mostly exhibit immune responses similar to wild-type. This correlation between non-activation of the immune genes and survival might be noteworthy for the future. A potential role of the communication between kynurenine and neuroinflammatory pathway (Mithaiwala et al, 2021) in the survival mechanism can be addressed in future investigation.

Finally, also the distinctive transcriptional patterns identified for SLC subfamilies can serve as an interesting starting point to identify critical metabolic rearrangements in the future, particularly in *letm1*-associated seizures, diabetes, and cancer diseases.

In summary, our work provides multiple perspectives on *letm1* pathogenicity that go beyond the field of mitochondrial cation homeostasis and may potentially change future clinical translation.

# Materials and Methods

## Zebrafish maintenance and husbandry

All animal research and husbandry was conducted according to Austrian and European guidelines for animal research (fish maintenance and care approved under BMWFW-66.006/0012-WF/II/3b/2014, experiments approved under BMWFW-66.006/0003-WF/V/3b/

2016, which is cross-checked by Geschäftsstelle der Kommission für Tierversuchsangelegenheiten gemäß § 36 TVG 2012 p. A. Veterinärmedizinische Universität Wien, A-1210, before being issued by the Bundesministerium für Bildung, Wissenschaft und Forschung). Zebrafish (*D. rerio*) were maintained under constant circulating water 26–28°C with a 16:8 white light/dark cycle. Husbandry and breeding followed standard protocols. Fish embryos were kept in the incubator at 28°C in E3 medium with a 16:8 white light/dark cycle.

## Generation of *letm1*-TALEN fish

### *Design and construction of letm1-TALENs*
Transcription activator-like effector nucleases (TALENs) was used to target the first exon of the zebrafish *letm1* gene (Ensembl gene ENSDART00000079558.7) according to Cermak et al (2011) and Doyle et al (2012). The design was made using TAL Effector Nucleotide Targeter 2.0 with a spacer width of 15–16 bp, RVD length of 15–20 bp, NN for G recognition, and Thymine as the upstream base. TAL effector modules were cloned into pFUS backbone vectors using the Golden Gate TALEN and TAL Effector kit (Addgene). The two arrays were joined in the final backbone vectors pCS2TAL3-DD or pCS2TAL3-RR including the homo-dimeric FokI nuclease domains. The following sequencing primers were used to check the expression vectors: Forward 5′-TTGGCGTCGGCAAA-CAGTGG-3′ and reverse 5′-CATCGCGCAATGCACTGAC-3′.

## Generation of TALEN mRNA and microinjection

The final TALEN construct was linearized with NotI and purified using the QIAGEN MinElute Reaction Cleanup kit and in vitro transcribed using Sp6 mMESSAGE mMACHINE kit (Thermo Fisher Scientific). Zebrafish embryos were injected with 300 ng/µl/pair of the TALEN mRNA, 0.3% TRITC-dextran (Sigma-Aldrich) in nuclease-free water. The target locus for genotyping was amplified using forward 5′-GCTCTATCTACAGGAAGACGGAAGCA-3′ and reverse 5′-TAACCCACACACTTATCTACCC-3′ primers.

## Genotyping of letm1-TALEN mutated fish

Injected specimens were genotyped at larval stage. Genomic DNA was isolated from fish lysed by incubating in 50 mM NaOH at 95°C for 15 min. The lysis was stopped with Tris–HCL (pH 8). The potential mutations were analyzed on agarose gel by size difference. DNA bands were gel extracted, sub-cloned into the pJET1.2 vector

showing the kynurenine pathway and Preiss–Handler and salvage pathways for NAD production, where certain enzymes are down-regulated in NC and C. NC: *letm1*–/– non-compensators and C: *letm1*–/– compensators. Dashed lines indicate intermediate metabolic steps. **(I)** Heat map of expression of enzymes involved in the different pathways of NAD production in pairwise comparisons C versus NC, C versus WT, and NC versus WT. Stars indicate the significance of the adjusted *P*-value obtained with the Benjamini–Hochberg method (\**P*-value < 0.05; \*\**P*-value < 0.01; \*\*\**P*-value < 0.001).

**Primers to amplify *letm1* fragment ligated into pGEM-T easy vector**

| Primers *letm1* coding sequence region | Sequence |
|---|---|
| Forward primer *letm1* | ATGGCATCGATTTTGCTCAC |
| Reverse primer *letm1* | ACAGAGCCACCAGCTGCGGC |

following the manufacturer's protocol Clone JET PCR Cloning kit (Thermo Fisher Scientific) and sequenced.

### RNA probes in situ hybridization

*letm1* digoxigenin-labeled riboprobe was synthesized from linearized plasmid pGEM-T easy containing *letm1* coding sequence. In vitro transcription was carried out at 37°C for 4 h using SP6 polymerase for the antisense probe and T7 for the sense probe. The RNA probe was purified using the RNA clean up kit (QIAGEN RNeasy kit), evaluated on agarose gel and stored in –80°C in hybridization mix (50% formamide, 5x SSC, 0.1% Tween-20, 0.5 mg/ml torula yeast RNA, and 50 μg/ml heparin).

### Whole-mount in situ hybridization

The whole-mount in situ hybridization was carried out on 1 and 6 dpf larvae. Larvae were collected at ZT23 and fixed using 4% PFA in PBS overnight at 4°C. Sense and antisense samples were equally treated. Samples were rehydrated by immersion in 75%, 50%, 25% methanol/PTW (PBS+0,1% Tween-20). Proteinase K (Merck) treatment was used for 5 min (1 dpf) and 30 min (6 dpf) followed by pre-hybridization in hybridization mix was carried out at 65°C for 2 h and samples were probed overnight at 65°C. Anti-DIG-Alkaline phosphatase-coupled antibody (Roche) was diluted 1:3,000 in 2% blocking reagent, overnight at 4°C. Samples were washed in MABT (100 mM maleic acid, 150 mM NaCl, H2O, pH 7.5) and nitro-blue tetrazolium (NBT) and 5-bromo-4-chloro-3′- indolyphosphate (BCIP) (Roche) diluted in SB-PVA was prepared to detect DIG-probes.

### Adult zebrafish brain/brain sectioning

Zebrafish adult brains were collected at ZT23 fixed in 4% PFA overnight at 4°C. The in situ hybridization steps before blocking are identical to the description above. Brains were mounted in 3% agarose/PBS and coronal brain slices were generated with a microtome (Leica VT1000S; Leica Biosystems). Slice size was 100 μm thick. Slices were blocked in 10% sheep serum for 2 h and incubated with Anti-DIG-Alkaline phosphatase-coupled antibody (1:3,000 dilution in 10% sheep serum) at 4°C overnight. Nitro-blue tetrazolium (NBT) and 5-bromo-4-chloro-3′- indolyphosphate (BCIP) (Roche) diluted in SB-PVA was prepared to detect DIG probes. The brain slices were mounted on slides and imaged using Nikon NIS elements.

### Subcellular localization studies in HeLa cells

CMV-driven Drletm1 C-terminally fused to eGFP was constructed using the gateway vector. HeLa cells (ATCC) were grown in Dulbecco's Modified Eagle Medium (Gibco), 10% FCS (Gibco), 100 U/ml penicillin, and 0.1 mg/ml streptomycin (Gibco) at 37°C and with 5% CO$_2$. Cells were seeded in *μ-Slide 8 Well* plates (#80821; Ibidi) and transfected using Turbofect following the manufacturer's instructions (Thermo Fisher Scientific TurboFect Transfection Reagent). Mitochondria were counterstained with 50 nM Mito-Tracker Red FM (Thermo Fisher Scientific) for 20 min at 37°C. Nuclei were stained with 1 μg/ml Hoechst (Thermo Fisher Scientific). Live cell imaging was performed using a 63× oil objective and the Zeiss LSM560 confocal microscope equipped with a 37°C chamber and a constant CO$_2$ flow (5%).

### Zebrafish cell culture

*letm1+/+* and *letm1–/–* cell lines were generated using 24 h post fertilization embryos according to Vallone et al (2007) and maintained in L15 (Leibovitz) culture medium with 15% fetal bovine serum and 100 U/ml penicillin/100 mg/ml streptomycin, 50 mg/ml gentamicin, at 27°C.

### Antibody production against Dr-Letm1

#### *Letm1: Max Perutz labs monoclonal antibody production produced in mice*

Anti-Letm1 monoclonal antibody was produced in mice by using a C-terminal fragment of zebrafish Letm1 (Protein ID: ENSDART00000079558.7. Location of the antigen: AA 462-768).

**Forward and reverse primer sequences used for qRT-PCR.**

| Gene | Forward primer (5′-3′) | Reverse primer (5′-3′) | Ensembl gene ID |
|---|---|---|---|
| *eef1a* | GTGGCTGGAGACAGCAAGA | AGAGATCTGACCAGGGTGGTT | NM_131263 |
| *letm1* | CCACCTTCGAGACACAGTCC | GGCCATCTCCAACTTCACTC | NM_001045208 |
| *nampt1* | ACAACGGTCACCTTCCCATC | ACCAGGCAAGTCTCTACCCA | XM_002661340 |
| *nampt2* | TGGCAAAGGCATGGATGTCT | CACAGTCACCAGCATGTCCT | NM_212668 |
| *period1b* | GTCGGATGATGACAAACAGC | CCAGAGACACAGACGGACCT | NM_212439 |
| *clock1a* | TCGGAAACTTTAAGTCCCTCAA | CACTCCCTCAAAGCCGTTT | NM_130957 |
| *period2* | CCAACGTGGACGAAGATGTA | GCAGCACCTTCTGGATGTCT | NM_182857 |
| *β-actin* | TCACTCCCCTTGTTCACAATAA | GGCAGCGATTTCCTCATC | NM_131031 |
| *letm2* | CCAGTGCCCAAACTGGAG | ACTGAGGAGGACAGGGGATT | XM_001339351 |
| *bmal1a* | TCCCAGAGCTCTGCAGATCT | CCGTGTCCATGCTATCGTCA | XM_005163022 |

**NAMPT phylogenetic tree.**

| Protein | Organism | Accession number |
| --- | --- | --- |
| NAMPT | *Homo sapiens* | ENSP00000222553 |
| NAMPT | *Mus Musculus* | ENSMUSP00000020886 |
| Nampt | *Petromyzon marinus* | ENSPMAT00000009291.1 |
| Namptb | *Latimeria chalumnae* | ENSLACT00000025212.1 |
| Namptb | *Lepisosteus oculatus* | ENSLOCP00000012733 |
| Nampt2 | *Danio rerio* | ENSDART00000037894.10 |
| NAmPRTase | *Oryzias latipes* | ENSORLP00000012822 |
| Namptb | *Xiphophorus maculatus* | ENSXMAP00000007822 |
| Namptb | *Gasterosteus aculeatus* | ENSGACT00000015385.1 |
| Nampt3.2 | *Danio rerio* | ENSDARG00000076694 |
| Nampta | *Lepisosteus oculatus* | ENSLOCP00000019495 |
| Nampt1 | *Danio rerio* | ENSDARP00000069804 |
| Nampt | *Oryzias latipes* | ENSORLP00000016533 |
| Nampta | *Gasterosteus aculeatus* | ENSGACT00000024960.1 |
| Nampta | *Takifugu rubripes* | ENSTRUT00000085858.1 |
| Nampta | *Tetraodon nigroviridis* | ENSTNIT00000021902.1 |
| Nampta | *Latimeria chalumnae* | ENSLACP00000005040 |
| Nampt | *Xenopus Laevis* | ENSXETT00000076483.1 |
| Nampta | *Anolis carolinensis* | ENSACAT00000014786.4 |
| Nampt | *Gallus gallus* | ENSGALT00000013144.5 |
| Nampt | *Sus scrofa* | ENSSSCP00000016366 |

**LETM1 phylogenetic tree.**

| Protein | Organism | Accession number |
| --- | --- | --- |
| Letm1 | *Danio rerio* | XP_009305455.1 |
| LETM1 | *Homo sapiens* | AAD13138.1 |
| LETM1 | *Mus musculus* | AAD13139.1 |
| LETM1 | *Rattus norvegicus* | NP_001005884.1 |
| Letm1 | *Oryzias latipes* | XP_023807733.1 |
| DmLETM1 | *Drosophila melanogaster* | AAM68316.1 |
| CeLETM1 | *Caenorhabditis elegans* | NP_506382.1 |
| Letm2 | *Danio rerio* | XP_001339387.1 |
| LETM2 | *Homo sapiens* | NP_001186588.1 |
| LETM2 | *Mus musculus* | NP_766600.2 |
| LETM2 | *Rattus norvegicus* | AAH87079.1 |
| Letm2 | *Oryzias latipes* | XP_004072229.1 |
| Mdm38p | *Saccharomyces cerevisiae* | NP_014615.1 |
| Mrs7p | *Saccharomyces cerevisiae* | NP_015450.1 |
| AtLETM1 | *Arabidopsis thaliana* | At3g59820 |
| AtLETM2 | *Arabidopsis thaliana* | AT1G65540.1 |
| HCCR1 | *Mus musculus* | AAH21361.1 |
| HCCR1 | *Rattus norvegicus* | AAI59427.1 |
| HCCR1 | *Homo sapiens* | AAH19274.1 |
| Hccr1 | *Danio rerio* | NP_001124088.1 |

## Immunofluorescence

The immunofluorescence was performed according to the protocol from Ikeda and Freeman (2019). Cells were seeded in the eight-well ibidi plate $5 \times 10^4$ per well. After 24 h, the cells were stained with 100- nM Mito-Tracker Red CMX Ros for 20 min in the dark at 28°C and were allowed to recover for 20 min. The cells were washed with PBS and fixed using 4% PFA/PBS for 20 min followed by incubation in neutralizing buffer (glycine PIPES), permeabilized with 0.2% Triton X-100 for 20 min, and refixed in methanol for 25 min at −20°C. Cells were blocked overnight at 4°C in 3% BSA in PBS containing 0.02% Triton. Antibody was diluted in the blocking solution (Letm1 1:200) and cells were incubated for 2 d at 4°C. After 3 × 5 min washing with 3% BSA/PBS-Triton, cells were incubated with the secondary antibody at room temperature for 2 h (AF488 1:2,000 diluted in 3% BSA/PBS-Triton). 3 × 5 min washing in PBS. Vecta shield mounting medium was used to mount the cells before imaging with the Zeiss LSM700 confocal microscope with the 63× oil objective.

## Immunoblot

Protein lysates from zebrafish larvae were prepared in a lysis buffer containing NP-40, with 2× cOmplete proteinase inhibitor (Roche). Samples were lysed using a plastic pestle and centrifuged 3 min at 17,000*g* at 4°C. Samples were stored at −80°C. Protein concentration was determined using Protein Assay Dye Reagent Concentrate from Bio-Rad

(Cat. no. 500-0006) and 30 μg of protein was loaded for each sample. Transfer was done using a 0.2-μm nitrocellulose membrane. Commercial antibodies used were actin (#A2066; Sigma-Aldrich) 1:5,000 in 2.5% non-fat dry milk and Tom20: Rabbit (#42406S; Cell Signaling) 1:1,000 in 5% non-fat dry milk. SuperSignal West Femto Maximum Sensitivity Substrate (Thermo Fisher Scientific) was used to detect chemiluminescence signal and ChemiDoc Imaging System from Bio-Rad for recording.

## NAD+/H measurements

Zebrafish embryos were collected at ZT5 and the yolk was removed using a deyolking buffer (1/2 Ginzburg Fish Ringer [1] without Calcium: 55 mM NaCl, 1.8 mM KCl, and 1.25 mM NaHCO$_3$). The procedure was performed according to a published protocol (Link et al, 2006). NAD measurement in 1 dpf zebrafish embryos was performed using NAD+/NADH-Glo Assay kit from Promega according to the manufacturer protocol. Standard curves were built using NAD (Cat. no. 8285; Sigma-Aldrich) and NADH (Cat. no. 6660; Sigma-Aldrich). Samples were lysed using PBS/Base/0.5% DTAB (Cat. no. D8638; Sigma-Aldrich). Protein concentration was measured using the BCA Assay.

## Zebrafish samples for qRT-PCR

Zebrafish embryos were kept in the incubator at 28°C under a 16-h light/8-h dark cycle. Samples were collected in 2.0 ml tubes, snap-frozen in liquid nitrogen, and stored at −80°C.

## RNA extraction and qRT-PCR

Larvae were euthanized in 15% ice-cold tricaine (MS-222) and decapitated. Heads and tails were independently pooled and snap-frozen in liquid nitrogen.

Anesthesia of the adult fish was performed in 30% ice-cold tricaine. The eyes, brain, liver, heart, swim-bladder, and gonads were snap-frozen in liquid nitrogen. Three organs were pooled into one biological replicate. 700 $\mu$l TRI reagent (Sigma-Aldrich) was added to the samples. Larvae were lysed for 2 min at 30 Hz using a metal bead (Peqlab Biotechnologie) and the QIAGEN tissue lyser. Total RNA was isolated using the Direct-Zol RNA MiniPrep (Zymo Research) according to the manufacturer's protocol. 200 and 400 ng of RNA were used to prepare cDNA with the QuantiTect reverse transcription kit (QIAGEN). qRT-PCR was performed using Luna Universal qRT-PCR Master Mix. The expression levels of the measured transcripts were normalized to $\beta$-actin. Primer sequences for *per1b*, *clock1a*, *per2*, and $\beta$-actin were described and tested previously (Dekens et al, 2017). Intron-spanning qRT-PCR primers for *nampt1* and *2* were designed using Primer3Plus.

## TEM

For TEM Zebrafish larvae at 4 dpf were fixed in 5% glutaraldehyde (Merck) in 0.1 M phosphate buffer (Sigma-Aldrich), pH 7.2, at 4°C for 3 h. Subsequently, samples were post-fixed in 1% osmium tetroxide (Merck) in the same buffer at 4°C for 2 h. After dehydration in an alcohol gradient series and propylene oxide (Merck), the tissue samples were embedded in glycid ether 100 (Serva). Ultrathin sections were cut on a Leica ultramicrotome (Leica Ultracut S), stained with uranyl acetate (Sigma-Aldrich), and lead citrate (Merck) and examined with a Zeiss TEM 900 electron microscope (Carl Zeiss) operated at 80 kV.

## RNA sequencing, differential expression analysis, and enrichment

Samples at 4 dpf larval stage were collected, frozen and stored at –80°C. RNA extraction was performed using the Direct-Zol RNA MiniPrep (Zymo Research) according to the manufacturer's protocol including the on-column gDNA digestion. Four biological replicates per condition were used. The Agilent RNA 6000 Nano kit (Agilent) was used to check the quality of the RNA samples. All samples were submitted to the VBCF NGS facility (Vienna) where library preparation, polyA enrichment, sequencing, alignment, and differential expression analysis were performed. Samples were sequenced on the Illumina NovaSeq platform with 150-base pair end reads. The data were aligned to the zebrafish genome using the Ensembl version GRCz10. *P*-values were obtained using the Wald test and adjusted *P*-values were calculated with the Benjamini-Hochberg method.

### Enrichment analysis and heat map

The functional enrichment analysis was performed using the online free access tool g:profiler (https://biit.cs.ut.ee/gprofiler/gost) with an adjusted *P*-value < 0.05 and a fold change of 1 as cut-offs. The enrichment map was build using the software Cytoscape v3.9.0. Heat maps were obtained using R (version 3.6.1).

## Phenotypical analysis

Zebrafish eggs were obtained from *letm1+/+* or *letm1−/−* pairs of fish and observed under a dissecting microscope. From 0 to 5 dpf, scores were kept according to different phenotypic categories: unfertilized, malformed, non-developing, dead, and malformed. Embryos were removed as soon as they were counted but malformed embryos were still counted in the statistics because they still survive for some time. Representative pictures were taken using the Nikon NIS Elements camera.

## Phylogenetic analysis

The phylogenetic analysis was calculated using *phylogeny.fr* (Dereeper et al, 2008) and the tree was visualized using itol.embl.de/.

# Data Availability

The RNA sequencing data from this publication have been deposited to the Dryad database and assigned with the identifier doi: 10.5061/dryad.jwstqjqbk.

# Supplementary Information

# Acknowledgements

We thank the members of the Tessmar-Raible, Nowikovsky, and Raible groups for discussions; Florian Raible for helping with the sequencing data; Andrej Belokurov, Margaryta Borysova, and Netsanet Berhane Getachew for excellent fish care at the MFPL aquatic facility; Tamara Bolcevic for screening of the anti-Dre-Letm1 sera for immunostaining; and the Next Generation Sequencing Facility at the Vienna BioCenter Core Facilities (VBCF) for the RNA sequencing. This work was supported by the Austrian Science Fund, Fonds zur Förderung der wissenschaftlichen Forschung (FWF) standalone grants P29077 and P31714 to K Nowikovsky, the Austrian Science Fund (FWF) grant DOC 72 doc.funds and I2972 to P Dao, the research platform "Rhythms of Life" of the University of Vienna and a FWF (http://www.fwf.ac.at/) SFB grant (#SFB F78), an FWF research project grant (#P28970), and funding from the European Research Council under the European Community's Horizon 2020 Programme ERC Grant Agreement 819952 to K Tessmar-Raible.

## Author Contributions

P Dao: formal analysis, investigation, visualization, methodology, and writing—original draft, review, and editing.
S Hajny: formal analysis, investigation, and methodology.
R Mekis: formal analysis, investigation, visualization, and methodology.
L Orel: methodology.
N Dinhopl: investigation, methodology, and writing—original draft.
K Tessmar-Raible: conceptualization, resources, supervision, funding acquisition, methodology, project administration, and writing—original draft, review, and editing.

K Nowikovsky: conceptualization, resources, supervision, funding acquisition, methodology, project administration, and writing—original draft, review, and editing.

## Conflict of Interest Statement

The authors declare that they have no conflict of interest.

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
