## [Reviewer comments · Life Science Alliance]

Life Science Alliance

The cation-exchanger Letm1, circadian rhythms and NAD(H) levels interconnect in diurnal zebrafish

Pauline Dao, Stefan Hajny, Ronald Mekis, Lukas Orel, Nora Dinhopf, Kristin Tessmar-Raible, and Karin Nowikovsky
DOI: <https://doi.org/10.26508/lsa.202101194>

Corresponding author(s): Karin Nowikovsky, University of Veterinary Medicine Vienna and Kristin Tessmar-Raible, University of Vienna

Review Timeline:

Submission Date:	2021-08-14
Editorial Decision:	2021-09-17
Revision Received:	2022-03-21
Editorial Decision:	2022-04-13
Revision Received:	2022-04-20
Accepted:	2022-04-21

Scientific Editor: Novella Guidi

Transaction Report:

September 17, 2021

Re: Life Science Alliance manuscript #LSA-2021-01194-T

Dr. Karin Nowikovsky
University of Veterinary Medicine Vienna
Biomedical Sciences; Physiology and Biophysics
Building HA
Veterinaerplatz 1
Wien 1210
Austria

Dear Dr. Nowikovsky,

Thank you for submitting your manuscript entitled "The cation-exchanger Letm1, circadian rhythms and NAD(H) levels interconnect in diurnal zebrafish" to Life Science Alliance. The manuscript was assessed by expert reviewers, whose comments are appended to this letter. As you will note from the reviewers' comments below, all the three reviewers are quite interested and raise few concerns. We, thus, encourage you to submit a revised version of the manuscript back to LSA that responds to all of the reviewers' points including extending the observational time window to corroborate the conclusion and run RT-PCR for the Letm1 to demonstrate the circadian expression of Letm1. Moreover, to explain the significant percentage of surviving animals in the letm1 KO, please measure the expression of other mitochondrial cation exchangers to check whether there is a compensatory mechanism, quantifying both cytoplasmic and mitochondrial calcium levels. In line with Reviewer 3, and to strengthen the conclusions, please conduct experiments showing changes in diurnal oscillations of seizure activity under Letm1 deficiency. All the other minor points should be addressed as well.

Thank you for this interesting contribution to Life Science Alliance. We are looking forward to receiving your revised manuscript.

Sincerely,

-- Summary blurb (enter in submission system): A short text summarizing in a single sentence the study (max. 200 characters including spaces). This text is used in conjunction with the titles of papers, hence should be informative and complementary to

the title and running title. It should describe the context and significance of the findings for a general readership; it should be written in the present tense and refer to the work in the third person. Author names should not be mentioned.

B. MANUSCRIPT ORGANIZATION AND FORMATTING:

Reviewer #1 (Comments to the Authors (Required)):

The manuscript "The cation-exchanger Letm1, circadian rhythms and NAD(H) levels interconnect in diurnal zebrafish" by Pauline Dao et al. describes the rather interesting link between Letm1, mitochondrial integrity/function and circadian rhythmicity in a very elegant way. It also provides the basis to better understand the pathophysiological mechanism behind the seizures in the Wolf Hirschhorn Syndrome.

I have only very minor comments:

- Authors seem to use Letm1 and LETM1 indistinctly, please homogenize within the text.
- The meaning of dpf, hpf, and TALEN is given until the methods section, but sentences including 6 dpf, TALEN technology, etc. appear before within the text. Please consider defining dpf, etc. where they are first mentioned.
- In the Immunoblot section, the name/brand of the device used to detect and record the chemiluminescence signal is missing.

Addendum:

Pauline Dao et al. have already done a good deal of experimental work, leading to a Letm1^{-/-} zebrafish model that might advance research on the role of Letm1.

Addressing the reviewers comments would certainly improve its quality.

Although Figs. 2i,j would suggest time of the day differences in Letm1 expression, I quite agree with reviewer #2 on the need to extend the time window.

All other comments by reviewer #2 and reviewer #3 should be addressed, in particular the ones asking for some explanation within the text.

As for experiments, further experiments are always needed, of course. For instance, after analysing the (compensatory?) expression of other mitochondrial cation exchangers, quantifying both cytoplasmic and mitochondrial calcium levels would have to be done...

Reviewer #2 (Comments to the Authors (Required)):

The study presented by P. Dao and co-workers aims to investigate the interconnection between circadian rhythms and the mitochondrial cation-exchanger Letm1. To this purpose a letm1 knock-out zebrafish model was established. A significant population of letm1 deficient animals were viable and on these the authors found deregulated NAD⁺/NADH metabolism and alteration in the circadian expression of NAMPT as well as of core clock genes.

The study is of interest but the linkage between Letm1 and the circadian clockwork might be indirectly due to dysregulation of the mitochondrial structure/function. Indeed, convincing evidences suggest an intimate interplay between clock genes and the mitochondrial energy metabolism so that any condition affecting the mitochondrial physiology will inevitably impact on circadian rhythms.

Specific points raised.

1. Figs. 2i,j: these are the only evidences in this study aimed to demonstrate the circadian expression of Letm1, however, the time points chosen do not show any daily rhythmic change. Rather, it appears as a progressive increase of Letm1. Moreover, in the supplementary Fig.1 the expression profile of Letm1 from ZT5 to ZT23 is quite different from that presented in the main text (i.e. with an apparent semicircadian period). The authors must strengthen this point and extend the observational time window to corroborate their conclusion. Furthermore, quantitative RT-PCR of the letm1 transcript would provide additional information and

complement the WB analysis.

2. Fig.3: the authors show the occurrence of a significant percentage of surviving animals in the *letm1* KO context but do not make any attempt to explain why. Is this due to a compensatory up-regulation in the expression of the ortholog *letm2* or of other mitochondrial cation exchangers? A quantitative RT-PCR would provide an answer.

3. Fig. 4b: the results shown in this figure (particularly for *letm1* *+/+* controls) conflict with the well-established notion that the steady state content of NAD⁺ is much higher than that of NADH; the authors are asked to explain such an anomaly. The concern raised at point 1 applies to the expression profiles shown in panels c-g; moreover the expression profile of the master clock gene *bmal1* is required.

Minor point.

The authors should mention that *Lemt1* is emerging as a pleiotropic protein that in addition to function as a cation exchanger appears to be involved in other mitochondrial functions such as shaping the IMM cristae morphology, tubular appearance of the organelle network, mitochondrial biogenesis.

Reviewer #3 (Comments to the Authors (Required)):

The work entitled "The cation-exchanger *Letm1*, circadian rhythms and NAD(H) levels interconnect in diurnal zebrafish" by Dao et al. and colleagues showed diurnal oscillations of protein levels of *Letm1* in wild-type zebrafish. While, *Letm1* was previously reported to be crucial for normal glucose metabolism, the present work shows for the first time that deficiency of *LETM1* reduces NAD⁺ and NADH pools. The authors suggest that these parameters could explain the Wolf Hirschhorn Syndrome (WHS)-associated epilepsy and sleep disorder. The manuscript is of interest, but additional experiments are required in order to support some claims and conclusions that are not fully supported by data.

1. The manuscript and its conclusion would take advantage from experiments showing the effect of *Letm1* deficiency on seizure activity in zebrafish. Especially, experiments showing changes in diurnal oscillations of seizure activity under *Letm1* deficiency.

2. Circadian oscillations of NAD⁺ levels have been previously reported (Nakahata et al., 2009), why diurnal rhythms of NAD⁺ levels in *letm1*^{-/-} fish are not measured throughout the current study?

3. The mitochondrial morphology of *letm1*^{-/-} larvae has been studied, however the implication of *Letm1* deficiency on mitochondrial function (respiration, ATP production, reactive oxygen species generation and transmembrane potential) should also be studied.

4. The authors claimed that *sirt1* and circadian clock gene regulation contribute to the increased diel amplitude changes of *nampt 1* and *nampt 2* transcripts in *letm1*^{-/-} fish, qPCR analyses showed an enhanced amplitude of all tested core circadian genes (*CLOCK*, *PER2*) in *letm1*^{-/-} fish. As it has been reported by Levine et al, 2020 that NAD⁺ enabled *CLOCK*/*BMAL* by *PER2* nuclear extrusion, we would expect a reduction in gene level of *CLOCK* in *letm1*^{-/-} fish as a result of NAD⁺ deficiency, however, an increase in *CLOCK* level has been reported. Please provide an explanation.

Reviewer #1 (Comments to the Authors (Required)):

The manuscript "The cation-exchanger Letm1, circadian rhythms and NAD(H) levels interconnect in diurnal zebrafish" by Pauline Dao et al. describes the rather interesting link between Letm1, mitochondrial integrity/function and circadian rhythmicity in a very elegant way. It also provides the basis to better understand the pathophysiological mechanism behind the seizures in the Wolf Hirschhorn Syndrome.

I have only very minor comments:

- Authors seem to use Letm1 and LETM1 indistinctly, please homogenize within the text.*
- The meaning of dpf, hpf, and TALEN is given until the methods section, but sentences including 6 dpf, TALEN technology, etc. appear before within the text. Please consider defining dpf, etc. where they are first mentioned.*
- In the Immunoblot section, the name/brand of the device used to detect and record the chemiluminescence signal is missing.*

We thank the reviewer for their positive evaluation. We are sorry for having missed the information on abbreviation when terms are first mentioned in the main text and included it now. The missing name/brand of the chemiluminescence detecting device is ChemiDoc™ MP, Imaging System from BIO-RAD and is now provided in the Method section: Immunoblot. We totally agree that the use of different nomenclatures for LETM1 is distracting, but we aimed to respect the standard nomenclature for mammals and zebrafish. We therefore used Letm1 when we mention the zebrafish protein and LETM1 when we mention the mammalian protein.

Addendum:

Pauline Dao et al. have already done a good deal of experimental work, leading to a Letm1-/- zebrafish model that might advance research on the role of Letm1.

Addressing the reviewers comments would certainly improve its quality.

Although Figs. 2I,J would suggest time of the day differences in Letm1 expression, I quite agree with reviewer #2 on the need to extend the time window.

We extended the time window for the Western blots and show that there is an even more stable diel regulation when an additional day of data is added.

All other comments by reviewer #2 and reviewer #3 should be addressed, in particular the ones asking for some explanation within the text.

As for experiments, further experiments are always needed, of course. For instance, after analysing the (compensatory?) expression of other mitochondrial cation exchangers, quantifying both cytoplasmic and mitochondrial calcium levels would have to be done...

In order to obtain an unbiased view on the compensatory mechanisms and also, because there are many (and probably in part undiscovered) other mitochondrial cation exchangers, we performed quantitative sequencing analyses. Interestingly, this provides highly novel perspectives onto the possible mechanisms relevant to compensate for the loss. In brief, compensation does not happen via other mitochondrial cation exchangers and we confirm our observations on the NAD⁺/NADH system. Unexpectedly, we observe a large regulation via the immune system between all the different comparisons. (Please keep in mind that the *letm1* compensated mutants are phenotypically normal.) While unexpected, it appears to be consistent with observation in other areas providing increasing evidence that the immune system is not just a defender to the outside, but also important for internal physiological balance. We believe that this insight provides significant additional scientific value to the manuscript.

However, considering that the compensators survived without altering Ca²⁺ transporter expression, quantification of cytoplasmic and mitochondrial Ca²⁺ levels is unlikely to provide further insight in direct connections to the conclusions of this particular study.

Reviewer #2 (Comments to the Authors (Required)):

The study presented by P. Dao and co-workers aims to investigate the interconnection between circadian rhythms and the mitochondrial cation-exchanger Letm1. To this purpose, a letm1 knock-out zebrafish model was established. A significant population of letm1 deficient animals were viable and on these the authors found deregulated NAD⁺/NADH metabolism and alteration in the circadian expression of NAMPT as well as of core clock genes.

The study is of interest but the linkage between Letm1 and the circadian clockwork might be indirectly due to dysregulation of the mitochondrial structure/function. Indeed, convincing evidences suggest an intimate interplay between clock genes and the mitochondrial energy metabolism so that any condition affecting the mitochondrial physiology will inevitably impact on circadian rhythms.

We thank Reviewer 2 for their encouraging comments and constructive suggestions.

Specific points raised.

1. Figs. 2I,J: these are the only evidences in this study aimed to demonstrate the circadian expression of Letm1, however, the time points chosen do not show any daily rhythmic change. Rather, it appears as a progressive increase of Letm1. Moreover, in the supplementary Fig.1 the expression profile of Letm1 from ZT5 to ZT23 is quite different from that presented in the main text (i.e. with an apparent semicircadian period). The authors must strengthen this point and extend the observational time window to corroborate their conclusion. Furthermore, quantitative RT-PCR of the letm1 transcript would provide additional information and complement the WB analysis.

This is a very good point. We addressed the reviewer's request and found that extending the observation time from 24 hours to 48 hours was indeed informative. The diel rhythm is even more robust at 7 dpf, further supporting the original conclusions of our study.

Following the reviewer's advice, we also analyzed the letm1 transcripts. At 6 dpf, at a time point that revealed the circadian transcription of clock genes, no transcriptional changes for *letm1* became apparent. Thus, diel oscillation occurs at the protein level through post-transcriptional/post-translational changes.

2. Fig. 3: the authors show the occurrence of a significant percentage of surviving animals in the *letm1* KO context but do not make any attempt to explain why. Is this due to a compensatory up-regulation in the expression of the ortholog *letm2* or of other mitochondrial cation exchangers? A quantitative RT-PCR would provide an answer.

We thank the reviewer for bringing to our attention the lack of explanation for the survival of *letm1*^{-/-} animals ("compensators"). As suggested, we included the expression analysis of *letm2*. However, qPCR data excluded its upregulation, which could have been implicated in a compensatory mechanism. We agree that other mitochondrial cation transporters also immediately come to mind as compensators. Because antibodies against zebrafish proteins are limited, and analysis of the transcription profile of mitochondrial cation transporters that may overlap with *letm1* function is complicated by their large number, numerous regulators and possible differences between human and zebrafish, we opted for a more global transcriptional analyses, which we have included in the revised manuscript. This provides several interesting and new insights. In brief, compensation does not happen via other mitochondrial cation exchangers and we confirm our observations on the NAD⁺/NADH system. Unexpectedly, we observe a large regulation via the immune system between all the different comparisons. (Please keep in mind that the *letm1* compensated mutants are phenotypically normal.) While unexpected, it appears to be consistent with observation in other areas providing increasing evidence that the immune system is not just a defender to the outside, but also important for internal physiological balance. We believe that this insight provides significant additional scientific value to the manuscript. The new analysis can be found in the result section: "Genetic compensations for *letm1* deficiency"

3. Fig. 4b: the results shown in this figure (particularly for *letm1* ^{+/+} controls) conflict with the well-established notion that the steady state content of NAD⁺ is much higher than that of NADH; the authors are asked to explain such an anomaly. The concern raised at point 1 applies to the expression profiles shown in panels C-G; moreover the expression profile of the master clock gene *bmal1* is required.

We thank the reviewer for pointing this out. Indeed, the wild-type NAD/H levels at 1 dpf are surprising, a possible explanation could be a glycolytic metabolic stage, because at this stage many non-differentiated, dividing cells exist. Those cells typically rely more on glycolysis than on oxphos metabolism. However, as there are no other studies on zebrafish that could serve as a reference, we acknowledge that the explanation at present remains a speculation. We discuss these conflicting data compared to mammals and adult organism as a comment to the results (p 8, lines 216-219). We also provided new qPCR data on *bmal1* in Fig.S3A.

Concerning the 48h graphs on circadian clock genes: the clock genes we used for analyses are well established for their diel/circadian profiles (Tamai *et al.*, 2004), including analyses over multiple developmental days. Our qPCR data are fully in line with these data. In order to point this out, we now add a reference to these previous analyses in the manuscript.

Minor point.

The authors should mention that Letm1 is emerging as a pleiotropic protein that in addition to function as a cation exchanger appears to be involved in other mitochondrial functions such as shaping the IMM cristae morphology, tubular appearance of the organelle network, mitochondrial biogenesis.

We are thankful for this suggestion and included the pleiotropic aspect of Letm1 functions in the introduction with the relevant references (p 3,4).

Reviewer #3 (Comments to the Authors (Required)):

The work entitled "The cation-exchanger Letm1, circadian rhythms and NAD(H) levels interconnect in diurnal zebrafish" by Dao et al. and colleagues showed diurnal oscillations of protein levels of Letm1 in wild-type zebrafish. While, Letm1 was previously reported to be crucial for normal glucose metabolism, the present work shows for the first time that deficiency of LETM1 reduces NAD+ and NADH pools. The authors suggest that these parameters could explain the Wolf Hirschhorn Syndrome (WHS)-associated epilepsy and

sleep disorder. The manuscript is of interest, but additional experiments are required in order to support some claims and conclusions that are not fully supported by data.

We thank the reviewer for their valuable comments.

1. The manuscript and its conclusion would take advantage from experiments showing the effect of Letm1 deficiency on seizure activity in zebrafish. Especially, experiments showing changes in diurnal oscillations of seizure activity under Letm1 deficiency.

As the Reviewer correctly mentioned above we suggest that our findings could explain letm1-related disorders of the WHS. This is a part of the discussion section, and also mentioned in the introduction to place our work in a more general context. Nowhere do we conclude anything in this direction based on our current results. Any useful scientific discussion should be able to speculate about the broader meaning of results. We would also like to point out that we do not have practical experience with required technology nor data analyses. Hence, as this clearly goes beyond the current direct conclusions we draw from our work and we do not have the experimental experience, we believe that this goes beyond what should be requested within the framework of a fair revision process.

We believe that the idea that Letm1 is not only responsible for seizures but also for a perturbed diurnal pattern is interesting and could be exploited in future research. Also, our new sequencing data, which may indicate that in addition to changes in the NAD⁺/NADH system, the immune system may be critical for survival decisions under Letm1 deficiency, may also provide fresh perspectives for future work. New phenotypic insights on the *letm1* deficiency have all the potential to elucidate the yet not understood mechanisms underlying the pathogenicity of *letm1*, especially that of seizures. However, it should be clear that this current work does not focus on seizures. Therefore, to study diurnal oscillations of seizure activity is beyond the focus of this study.

2. Circadian oscillations of NAD⁺ levels have been previously reported (Nakahata et al., 2009), why diurnal rhythms of NAD⁺ levels in letm1^{-/-} fish are not measured throughout the current study?

While the question whether NAD⁺ levels are not only different but also oscillate differently in absence of *letm1* is interesting, this goes beyond the technical feasibility and is not required to support the main conclusions of our manuscript, as we do not claim that NAD⁺ level oscillate. They may or they may not- this is an interesting aspect for future work, triggered by the interesting results of our manuscript.

Our measurements are done using 5-6 biological replicates with each n=20 larvae, for *letm1+/+* and *letm1-/-*. We used 1 dpf, a timepoint before clocks start ticking. For a question in the context of diel regulation, we would choose larvae at 6 and 7 dpf, as this is also when we know that *Letm1* levels are clearly oscillating.

At this timepoint we could only compare *letm1+/+* and *letm1-/-* compensators (since all other *letm1-/-* do not survive 4 dpf). We would need 100-120 larvae per genotype/phenotype, and this for at least (!) 4 time points each per 24 hours. As for a test for a stable oscillation, we should monitor at least 48h, this number doubles.

This completely exceeds the number of *letm1-/-* compensator progeny that can realistically be obtained.

3. The mitochondrial morphology of letm1-/- larvae has been studied, however the implication of Letm1 deficiency on mitochondrial function (respiration, ATP production, reactive oxygen species generation and transmembrane potential) should also be studied.

We agree with the reviewer that mitochondrial functional deficiencies in oxidative ATP production, ROS generation or membrane potential were not assessed. As we present for the first time a *letm1* study in the zebrafish model system, it would be justified to include ATP, ROS, and other bioenergetics data. However, we are using zebrafish, an excellent diurnal animal model, to open new aspects of *Letm1* rather than to recapitulate what has been shown in mammals, flies or worms, which in the frame of this particular work may become distracting to the reader. We believe that the absence of such classical bioenergetics data in this particular study does not diminish the value of our new findings.

4. The authors claimed that sirt1 and circadian clock gene regulation contribute to the increased diel amplitude changes of nampt 1 and nampt 2 transcripts in letm1-/- fish, qPCR analyses showed an enhanced amplitude of all tested core circadian genes (CLOCK, PER2) in letm1-/- fish. As it has been reported by Levine et al, 2020 that NAD + enabled CLOCK/BMAL

by PER2 nuclear extrusion, we would expect a reduction in gene level of CLOCK in letm1-/- fish as a result of NAD+ deficiency, however, an increase in CLOCK level has been reported. Please provide an explanation.

We thank the reviewer for this valuable comment.

We note that the work of Levine was conducted in mice at the age of 8 months. The correlation of NAD+ and gene expression was investigated at different time points over the day/night cycle. Our data are unfortunately not from young adult zebrafish. Besides not using the same organism and age, we believe that NADH changes are prevailing NAD+, so that a direct effect of NAD+ on gene expression may possibly not be drawn.

April 13, 2022

RE: Life Science Alliance Manuscript #LSA-2021-01194-TR

Dr. Karin Nowikovsky
University of Veterinary Medicine Vienna
Biomedical Sciences; Physiology and Biophysics
Building HA
Veterinaerplatz 1
Wien 1210
Austria

Dear Dr. Nowikovsky,

Thank you for submitting your revised manuscript entitled "The cation-exchanger Letm1, circadian rhythms and NAD(H) levels interconnect in diurnal zebrafish". We would be happy to publish your paper in Life Science Alliance pending final revisions necessary to meet our formatting guidelines.

- please upload your main manuscript text as an editable doc file;
- Please upload all figure files as individual ones, including the supplementary figure files; all figure legends should only appear in the main manuscript file
- please add ORCID ID for secondary corresponding author-she should have received instructions on how to do so
- please add the Twitter handle of your host institute/organization as well as your own or/and one of the authors in our system
- please add your main and supplementary figure legends to the main manuscript text after the references section
- we encourage you to revise the figure legends for figures 4, 5 such that the figure panels are introduced in an alphabetical order
- please revise panels in figure 5 and also its callouts in the manuscript text
- please use the [10 author names, et al.] format in your references (i.e. limit the author names to the first 10)
- please be sure to use Capital Letters when introducing the figure callouts in the manuscript text
- please add callouts for Figures 1H; 4C, D; S1A-D; S3B and S5A-B to your main manuscript text
- please add a Data Availability section where you would include your RNA seq data deposition

A. FINAL FILES:

B. MANUSCRIPT ORGANIZATION AND FORMATTING:

Sincerely,

Reviewer #2 (Comments to the Authors (Required)):

The authors fulfilled satisfactory all the points raised by this reviewer with convincing additional experimental evidence. Therefore, the manuscript in the revised form has improved in soundness providing significant advance in the field and its acceptance for publication strongly recommended.

Reviewer #3 (Comments to the Authors (Required)):

The work entitled "The cation-exchanger Letm1, circadian rhythms and NAD(H) levels interconnect in diurnal zebrafish" by Dao et al. and colleagues showed diurnal oscillations of protein levels of Letm1 in wild-type zebrafish. While, Letm1 was previously reported to be crucial for normal glucose metabolism, the present work shows for the first time that deficiency of LETM1 reduces NAD⁺ and NADH pools. The authors suggest that these parameters could explain the Wolf Hirschhorn Syndrome (WHS)-associated epilepsy and sleep disorder. The manuscript is of interest, but additional experiments were requested in order to support some claims and conclusions that are not fully supported by data. For example:

1. experiments showing changes in diurnal oscillations of seizure activity under Letm1 deficiency and how this could be correlated with NAD⁺ and NADH pools would ultimately clarify the role of these parameters in epilepsy as the authors claimed. The authors responded by claiming that some of the suggested experiments are either beyond the focus of the manuscript.
2. The mitochondrial morphology of letm1^{-/-} larvae has been studied, additional experiments showing the implication of Letm1 deficiency on mitochondrial function were suggested, the authors claimed that these experiments will not add to the value of the manuscript.

April 21, 2022

RE: Life Science Alliance Manuscript #LSA-2021-01194-TRR

Dr. Karin Nowikovsky
University of Veterinary Medicine Vienna
Biomedical Sciences; Physiology and Biophysics
Building HA
Veterinaerplatz 1
Wien 1210
Austria

Dear Dr. Nowikovsky,

Thank you for submitting your Research Article entitled "The cation-exchanger Letm1, circadian rhythms and NAD(H) levels interconnect in diurnal zebrafish". It is a pleasure to let you know that your manuscript is now accepted for publication in Life Science Alliance. Congratulations on this interesting work.

DISTRIBUTION OF MATERIALS:

Again, congratulations on a very nice paper. I hope you found the review process to be constructive and are pleased with how the manuscript was handled editorially. We look forward to future exciting submissions from your lab.

Sincerely,
